# Polycystin-1 Is a Crucial Regulator of BIN1 Expression and T-Tubule Remodeling Associated with the Development of Dilated Cardiomyopathy

**DOI:** 10.3390/ijms24010667

**Published:** 2022-12-30

**Authors:** Magda C. Díaz-Vesga, Raúl Flores-Vergara, Jaime A. Riquelme, Marcelo Llancaqueo, Gina Sánchez, Cecilia Vergara, Luis Michea, Paulina Donoso, Andrew F. G. Quest, Ivonne Olmedo, Zully Pedrozo

**Affiliations:** 1Programa de Fisiología y Biofísica, ICBM, Facultad de Medicina, Universidad de Chile, Santiago 8380453, Chile; 2Advanced Center for Chronic Diseases, Facultad de Medicina, Universidad de Chile, Santiago 8380453, Chile; 3Departamento de Ciencias Básicas de la Salud, Facultad de Ciencias de la Salud, Pontificia Universidad Javeriana Cali, Cali 760031, Colombia; 4Advanced Center for Chronic Diseases, Facultad de Ciencias Químicas y Farmacéuticas, Universidad de Chile, Santiago 8380492, Chile; 5Departamento de Química Farmacológica y Toxicológica, Facultad de Ciencias Químicas y Farmacéuticas, Universidad de Chile, Santiago 8380492, Chile; 6Departamento Cardiovascular, Hospital Clínico Universidad de Chile, Santiago 8380456, Chile; 7Programa de Fisiopatología, ICBM, Facultad de Medicina, Universidad de Chile, Santiago 8380453, Chile; 8Centro de Estudios en Ejercicio, Metabolismo y Cáncer (CEMC), ICBM, Facultad de Medicina, Universidad de Chile, Santiago 8380453, Chile; 9Departamento de Biología, Facultad de Ciencias, Universidad de Chile, Santiago 7750000, Chile; 10Division of Nephrology, Department of Medicine, Hospital Clínico Universidad de Chile, Santiago 8380456, Chile; 11Laboratorio de Comunicaciones Celulares, Programa de Biología Celular y Molecular, ICBM, Facultad de Medicina, Universidad de Chile, Santiago 8380453, Chile

**Keywords:** BIN1, polycystin-1, dilated cardiomyopathy, T-tubule, heart failure

## Abstract

Cardiomyopathy is commonly observed in patients with autosomal dominant polycystic kidney disease (ADPKD), even when they have normal renal function and arterial pressure. The role of cardiomyocyte polycystin-1 (PC1) in cardiovascular pathophysiology remains unknown. PC1 is a potential regulator of BIN1 that maintains T-tubule structure, and alterations in BIN1 expression induce cardiac pathologies. We used a cardiomyocyte-specific PC1-silenced (PC1-KO) mouse model to explore the relevance of cardiomyocyte PC1 in the development of heart failure (HF), considering reduced BIN1 expression induced T-tubule remodeling as a potential mechanism. PC1-KO mice exhibited an impairment of cardiac function, as measured by echocardiography, but no signs of HF until 7–9 months of age. Of the PC1-KO mice, 43% died suddenly at 7 months of age, and 100% died after 9 months with dilated cardiomyopathy. Total BIN1 mRNA, protein levels, and its localization in plasma membrane-enriched fractions decreased in PC1-KO mice. Moreover, the BIN1 + 13 isoform decreased while the BIN1 + 13 + 17 isoform was overexpressed in mice without signs of HF. However, BIN1 + 13 + 17 overexpression was not observed in mice with HF. T-tubule remodeling and BIN1 score measured in plasma samples were associated with decreased PC1-BIN1 expression and HF development. Our results show that decreased PC1 expression in cardiomyocytes induces dilated cardiomyopathy associated with diminished BIN1 expression and T-tubule remodeling. In conclusion, positive modulation of BIN1 expression by PC1 suggests a novel pathway that may be relevant to understanding the pathophysiological mechanisms leading to cardiomyopathy in ADPKD patients.

## 1. Introduction

Autosomal dominant polycystic kidney disease (ADPKD) is an inherited kidney disease caused by mutations in polycystic kidney disease 1 gene (*PKD1*) and/or *PKD2* [1,2], whereby *PKD1* mutations lead to the most severe ADPKD manifestations [3,4]. Cardiovascular disorders are the most common cause of mortality in ADPKD patients, and the concept of polycystic kidney disease cardiomyopathy has been recently suggested [1,5,6,7]. Indeed, ADPKD was related to idiopathic dilated cardiomyopathy (DCM) development [6,7,8]. However, despite such insight, knowledge about the molecular and cellular mechanisms underlying the cardiac pathology involving *PKD1*/*PKD2* is scarce.

Polycystin dysfunction in cardiovascular cells may contribute to cardiac consequences of ADPKD. The *PKD1* and *PKD2* genes encode polycystin-1 (PC1) and polycystin-2, respectively, which are members of a family of transmembrane proteins expressed in several cells, including renal epithelial cells and cardiac cells [9]. 

Previous reports in young mice (9–11 weeks old) with PC1 selectively silenced in cardiomyocytes (PC1-KO) showed systolic and diastolic dysfunction, reduced protein levels of the L-type calcium channel (Cav1.2), reduced calcium transients [10], and diminished duration of cardiomyocyte action potentials [11], without any signs of distress, hypertrophy, or heart failure (HF). These results showed that PC1 expressed in cardiomyocytes is necessary to maintain normal heart function and calcium homeostasis [10,11]; however, the mechanism by which PC1 modulates cardiac function is unknown. 

The membrane scaffolding protein bridging integrator-1 or amphiphysin-2 (BIN1) is a cardiac transverse tubule protein, responsible for T-tubule membrane microdomain formation and calcium handling essential for normal beat-to-beat contractility [12]. BIN1, a member of the BIN-amphiphysin/Rvs (BAR) domain-containing protein superfamily [13], is encoded by a gene with 20 exons that are selectively spliced to produce the several isoforms expressed in different tissues [12,14]. Four different isoforms are expressed in cardiac tissue: a small constitutive BIN1 (BIN1 + 12), a constitutively alternative spliced BIN1 + 17, and two cardiac alternatively spliced variants: BIN1 + 13 and BIN1 + 13 + 17 [12,15]. 

Dimers of BIN1 insert into the membrane through their N-terminal domain, causing an invagination that initiates the formation of a tubule [14,16]. Moreover, BIN1 + 13 and BIN1 + 13 + 17 play a key role in the formation, abundance, and functionality of T-tubules, [12,14] as well as of microfolds, microdomains that support cardiac dyad formation [13,14,15]. Cardiac conditional BIN1-deleted adult mice show loss of T-tubule microfolds, increases in T-tubule luminal diameter, impaired cardiac excitation–contraction coupling [12] and cardiac contractility, reduced Cav1.2 protein levels [17], and over time develop dilated cardiomyopathy [18]. In fact, BIN1 is downregulated, and changes in the cardiomyocyte T-tubule ultrastructure have been reported to precede HF [17,19,20]. Moreover, decreased levels of BIN1 in plasma have been related to HF development in humans [21,22,23], suggesting it may have potential as a new biomarker; however, the pathway that regulates BIN1 expression remains undefined. 

Since T-tubules are crucial to maintaining cardiac contractility and BIN1 is a key protein for tubulogenesis, we hypothesize that PC1 regulates the content of BIN1 in the cardiac tissue, preserving T-tubule structure and cardiac function. In this study, we evaluated the relevance of PC1 in regulating BIN1 isoform expression and their relevance in HF development using a cardiomyocyte-specific PC1-KO mouse model. 

Considering that ADPKD patients develop cardiomyopathy and that cardiospecific PC1-KO induces cardiac dysfunction in a mouse model [10], we focused on determining whether BIN1 expression and T-tubule formation are dependent on PC1. This knowledge may help to understand the mechanism by which PC1 regulates cardiac function and leads to the development of cardiomyopathy in ADPK patients. 

## 2. Results

### 2.1. Sudden Death in Dilated Cardiomyopathy Is Associated with Loss of Polycystin-1 Expression

Cardiomyocyte-specific PC1-knockout (PC1-KO) in young mice (9–11 weeks old), decreased cardiac function, but did not lead to changes in the heart rate, cardiac hypertrophy or signs of HF or distress [10,11]. Thus, we studied the impact of the loss of cardiomyocyte PC1 expression on HF progression and the lifespan of adult mice. 

PC1-KO mice developed normally up to 7 months. In the following 2 months, they showed signs of dyspnea of abrupt initiation, shallow breathing, restricted mobility, and fatigue. One to two days after the appearance of these signs, the animals died. About 43% of PC1-KO mice died suddenly at 7 months of age and the rest within the following two months. The survival curve is shown in Figure 1A.

We evaluated cardiac function by echocardiography in mice without (<7-month-old mice) and with (7–9-month-old mice) signs of HF. Our data (Figure 1B) revealed a decreased ejection fraction in <7-month-old mice (EF: 72.4 ± 4.3% for control versus 56.7 ± 3.9% for PC1-KO) and 7–9-month-old mice (EF: 69 ± 3.7% for control versus 42 ± 4.0% for PC1-KO). Moreover, fractional shortening was detected in <7-month-old mice (FS: 38.6 ± 2.5% for control versus 25.3 ± 2.3 for PC1-KO) and 7–9-month-old mice (FS: 33.3 ± 2.5% for control versus 18 ± 1.6% for PC1-KO) (Figure 1C). We also evaluated cardiac morphology, morphometric, and molecular parameters as soon as the first signs of distress appeared in the PC1-KO mice. Representative hearts stained with hematoxylin/eosin are shown (Figure 1D). As shown, <7-month-old hearts from PC1^F/F^ and PC1-KO appear to be similar, but cardiac chambers are clearly dilated in 7–9-month-old PC1-KO mice. Chamber dilatation was confirmed by echocardiography (Appendix A). We observed increased left ventricular end-systolic (LVEsD) and diastolic (LVEdD) dimensions in 7–9-month-old PC1-KO mice (LVEsD: 1.78 ± 0.18 mm for control versus 3.15 ± 0.4 mm for PC1-KO, and LVEdD: 2.67 ± 0.16 mm for control versus 3.75 ± 0.39 mm for PC1-KO), while PC1-KO mice <7-month-old did not show significant differences compared to controls. Heart weight/tibia length (HW/TL) increased from 10.7 ± 0.6 to 17.2 ± 1.1 mg/mm (60%) in 7–9-month-old mice, without noticeable changes in <7-month-old mice (Figure 1E). Increases in heart weight were associated with lung wet weight/tibia length (LWW/TL) and varied from 11.3 ± 0.5 to 15.2 ± 0.7 mg/mm (35%) in 7–9-month-old mice (Figure 1F). Levels of mRNA expression increased in the hearts of 7–9-month-old PC1-KO mice for β-MHC, BNP, TGF-β, CTGF, and collagen 1 (Col1), used here as markers of cardiac stress and fibrosis (Appendix A). In addition, Masson’s trichrome staining of PC1^F/F^ and PC1-KO hearts did not reveal any differences in collagen deposits (Appendix A). Together, these data suggest that cardiomyocyte-specific PC1 deficiency induces cardiac dysfunction that progresses to cardiomyopathy and sudden death over time. Additionally, we did not detect changes in hypertrophy markers prior to the development of HF signs, coincident with observations described in previous reports [10].

### 2.2. Polycystin-1 Regulates Cardiac BIN1 Expression

As reduced BIN1 expression has been linked to HF development [22], we evaluated whether PC1 regulates cardiac BIN1 protein content in ventricular cardiac tissue. Our results did not reveal differences in BIN1 protein content between 7-day-old PC1^F/F^ and PC1-KO mice (Appendix A); however, total BIN1 protein levels decreased to around 50% of control levels in <7-month-old and 7–9-month-old PC1-KO mice, as compared to PC1^F/F^ mice (Figure 2A,B). Similar immunoblot results were obtained using a second BIN1 antibody (ab95022) that recognizes a different BIN1 amino acid sequence (Appendix A). Moreover, decreased BIN1 protein levels in cardiac tissue were associated with diminished BIN1 total mRNA content in PC1-KO mice, and detected even before any signs of HF became apparent (Figure 2C,D). Taken together, these results suggest that PC1 is critical to maintaining the expression of BIN1 protein in cardiomyocytes. 

Interestingly, the evaluation of changes in BIN1 protein levels over time in cardiac tissue revealed increases in the level of this protein associated with age in PC1^F/F^ mice. On the contrary, the increases in BIN1 protein with age were not observed in the PC1-KO mice (Appendix A).

Microsomal fractions enriched in plasma membrane components (MS) from PC1-KO and PC1^F/F^ mice were used to study BIN1 membrane localization. Beyond the decrease in total protein levels in PC-1-KO heart tissue, BIN1 presence in the membrane fractions of <7-month-old PC1-KO mice (Figure 2E) and 7–9-month-old PC1-KO mice (Figure 2F), was also decreased, as compared to those from PC1^F/F^ mice. 

Together, these data indicate that the absence of PC1 expression in cardiomyocytes leads to a decrease in BIN1 expression and also membrane localization, and that these changes can be detected even before any signs of distress typical of HF are observed. These results point towards PC1 as a previously unrecognized regulator of BIN1 expression in cardiac tissue. 

### 2.3. Cardiac BIN1 Isoform Expression Is Regulated by Polycystin-1

As total BIN1 mRNA decreased in PC1-KO mice, we evaluated whether PC1 caused isoform-specific changes in BIN1 expression by measuring BIN1 splice variant mRNA levels in the cardiac tissue of PC1-KO mice before (<7 months old) and after (7–9 months old) signs of distress appeared. As shown (Figure 3A,B), the expression of cardiac-specific BIN1 + 13 isoform decreased in PC1-KO mouse cardiac tissue, even before any signs of stress were detectable, while surprisingly, the relative abundance of the cardiac-specific BIN1 + 13 + 17 isoform increased in the cardiac tissue of PC1-KO mice (Figure 3C), as compared to control mice. The observed increase in BIN1 + 13 + 17 isoform presence in cardiac tissue of PC1-KO mice was transient, given that for PC1-KO mice with clear signs of HF the levels of BIN1 + 13 + 17 isoform decreased compared to control mice (Figure 3D). For both of the ubiquitous BIN1 isoforms, BIN1 + 17 and small BIN1, protein levels did not change before mice showed signs of distress; however, their levels decreased notably in the heart tissue of PC1-KO mice with HF signs (Appendix A). 

Together, these data suggest that decreased BIN1 total mRNA is mainly due to the loss of BIN1 + 13 isoform transcripts. Alternatively, the transient predominance of the BIN1 + 13 + 17 isoform in PC1-KO mice may reflect the consequence of a mechanism to counteract the onset of HF. To evaluate whether changes in BIN1 isoforms are the direct consequence of decreased PC1 expression, we used NRVMs in which PC1 was knocked down (Appendix A). Our results show that while BIN1 + 13 mRNA decreased, BIN1 + 13 + 17 mRNA increased (Appendix A). Moreover, ubiquitous BIN1 (BIN1 + 17 and small BIN1) isoforms did not change their mRNA levels in siPC1 NRVMs compared to controls (Appendix A), supporting our previous findings in <7-month-old PC1-KO mice and suggesting that BIN1 expression depends directly on PC1 expression in cardiomyocytes. Furthermore, the overexpression of PC1 in NRVM increases all BIN1 mRNA isoforms except BIN1 + 13 + 17 mRNA which decreases (Appendix A)

### 2.4. Ultrastructure of T-Tubules Correlates with Changes in Cardiomyocyte Polycystin-1 Expression

The ultrastructure and T-tubule formation are known to depend on cardiac BIN1 isoform expression and changes in both are related to HF onset [15]. Therefore, we studied the ultrastructure of T-tubules in the hearts of PC1-KO mice using transmission electron microscopy (TEM), and representative TEM images are shown (Figure 4A). Our observations indicate that lumen area increased (Figure 4B), while intraluminal electron density (as a measure of microfolds) of T-tubules in cardiomyocytes from PC1-KO cardiac tissue decreased (Figure 4C), and did so even before the appearance of distress signs (<7-month-old mice). These changes were maintained in 7–9-month-old PC1-KO mouse cardiomyocytes, as compared with those of PC1^F/F^ mice (Figure 4D,E). Interestingly, both the lumen area and intraluminal electron density increased and decreased, respectively, upon comparing cardiomyocyte T-tubules from <7-month-old and 7–9-month-old PC1-KO mice (Figure 4F,G), and these changes correlated with HF progression. Moreover, while T-tubule number per area did not change in the hearts of <7-month-old PC1-KO mice, decreased T-tubule numbers per area were observed in 7–9-month-old PC1-KO mice, as compared to PC1^F/F^ mice (Figure 4H,I). 

Overall, these data suggest that maintenance of cardiomyocyte T-tubule ultrastructure depends, at least in part, on PC1 expression. Moreover, altered T-tubule ultrastructure is related to dysfunction of cardiac contraction, as measured by echocardiography in PC1-KO mice, and these changes were detectable even before observing any signs of HF (compare Figure 4B,C with Figure 1B,C).

### 2.5. Plasma BIN1 + 13 + 17 Levels Correlate with Its Cardiac Expression in Cardiomyocytes Polycystin-1 Knockout Mice

Cardiac BIN1 + 13 + 17 (cBIN1) isoform levels in blood plasma are proportional to the levels of expression in cardiomyocytes [14] and the cBIN1 score (CS) is inversely proportional to blood plasma BIN1 + 13 + 17 content, suggesting it may serve as a new biomarker for early detection of HF [21,22]. However, information about cBIN1 score prior to signs of heart failure is lacking. Since we found an increase in BIN1 + 13 + 17 mRNA in heart tissue of PC1-KO mice, before noticing any signs of HF, we decided to determine CS in PC1-KO mouse blood plasma at this stage of disease evolution (<7-month-old mice). Interestingly, a decreased CS (Figure 5A) was observed in KO plasma compared with values observed for PC1^F/F^ mice. These observations support the notion that although the absence of PC1 is associated with decreased expression of total BIN1, BIN1 + 13 + 17 isoform levels are increased. In contrast and consistent with isoforms mRNA levels, BIN1 score was increased in PC1-KO mice with HF signs (Figure 5B).

According to these results, any change in the BIN1 + 13 + 17 plasma levels and CS with respect to the controls could be related to cardiac dysfunction and may serve as a biomarker of cardiac dysfunction prior to overt HF. 

Taken together, these data indicate that PC1 is a critical mechanosensor that maintains basal cBIN1 expression in cardiomyocytes essential for T-tubule formation, preservation of tissue ultrastructure, and cardiac function. Moreover, deregulation of PC1 expression induces cardiac dysfunction by decreasing the expression of some cBIN1 isoforms, which lead to T-tubule alterations that over time evolve into dilated cardiomyopathy, heart failure, and sudden death. In Figure 6, the most important findings of this study are summarized in a graphical model.

## 3. Discussion

Many ADPKD patients develop cardiomyopathy despite normal renal function and arterial pressure, suggesting that cardiovascular pathologies could be the result of polycystin mutations and dysfunction in heart and vascular tissue, rather than the consequence of renal dysfunction. However, this issue remains unclear and contradictory [5,6,24,25,26].

Interestingly, in a cohort of ADPKD patients, 6% of the individuals developed idiopathic dilated cardiomyopathy (DCM), while only 2.5% developed hypertrophic obstructive cardiomyopathy [6,26]. Our results show for the first time in a preclinical mouse model that altered expression of polycystin-1 (PC1), specifically in cardiomyocytes, is associated directly with DCM, heart failure development, and sudden death related to age. Moreover, although PC1-KO mice show a decreased ejection fraction, as measured by echocardiography, from 9 weeks of age onwards [10], they do not show any signs of stress until around when they are 7 months old. At that point, 43% of the PC1-KO mice died suddenly, and by 9 months, all died with evident signs of heart failure. Considering the differences in mouse and human lifespan [27], the first signs of heart failure extrapolated to human years would be equivalent approximately to 23–30 years of age. Indeed, clinical symptoms and signs of cardiovascular complications in ADPKD patients, such as hypertension, have been described for individuals of 27 years [26] and 32–34 years of age [28]. Differences in clinical symptoms, temporal onset, and severity for patients compared with our mouse model may be attributable, at least in part, to the fact that the cardiomyocyte-specific knockout affects both *Pkd1* alleles, while ADPKD patients have heterozygous mutations. On the other hand, a case of polycystic kidney disease was described for a girl of 8 months of age, who suddenly developed DCM and died 3 h after the symptoms appeared [7]. In agreement, our observations in the mouse knockout revealed that reductions in PC1 expression or function in cardiomyocytes are highly detrimental and promote the development of HF. 

T-tubule disruption, loss, and remodeling have been reported as crucial events that contribute to the development of heart failure [20,29,30,31,32]; however, the pathways that regulate tubulogenesis are not completely understood. Amphyphysin-2 or bridging integrator-1 (BIN1) is involved in tubulogenesis and T-tubule ultrastructure formation, both essential for maintaining cardiac function [33,34]. Indeed, cardiac-specific deletion of BIN1 induces age-associated DCM, starting at 8–10 months of age [18], and in vivo overexpression of exogenous cBIN1 reduced myocardial remodeling and dysfunction in a pathological hypertrophy model induced by isoproterenol [35]. Others showed that BIN1 preserves T-tubule and especially microfold structure, thereby avoiding the development of HF [12,14,36]. Our results show that PC1 is required for BIN1 expression in the heart tissue, and decreased PC1 expression reduced BIN1 expression and altered localization to the plasma membrane. Interestingly, our results suggest that BIN1 expression increases throughout the lifespan of mice, and this increment may be relevant to prolonged animal well-being. This may explain why decreased expression or function of PC1 predisposes animals more to cardiomyopathies as they age. 

On the other hand, two different cardiac BIN1 isoforms have been described as being relevant to T-tubule formation and ultrastructure [12,15]. While BIN1 + 13 is thought to represent the predominant isoform in cardiac tissue important for T-tubule formation, BIN1 + 13 + 17 is more implicated in microfold maintenance [12,13]. Although total BIN1 and BIN1 + 13 mRNA levels were lower in the heart-tissue-specific PC1-KO mice with decreased ejection fraction, but without any signs of HF, BIN1 + 13 + 17 mRNA levels were found to be elevated. Interestingly, increased levels of this isoform are no longer observed later on. Conversely, it was associated with the appearance of signs of DCM, which suggest that increased BIN1 + 13 + 17 levels may initially prevent detrimental T-tubule remodeling and the occurrence of HF. Furthermore, loss of BIN1 + 13 + 17 overexpression in the cardiac tissue of mice with signs of HF has been correlated with a greater decrease in the number of microfolds and T-tubules per unit area. These data are in agreement with previous reports showing that overexpression of this BIN1 isoform suffices to avoid T-tubule remodeling and cardiac dysfunction [37]. Moreover, increases in T-tubule lumen area and decreases in the number of microfolds were found to relate to cardiac dysfunction due to loss of T-tubule fuzzy space required to maintain the basal ionic flux [12]. This observation may explain, at least in part, the dysfunction in cardiac contractility observed in PC1-KO mice without signs of HF. 

BIN1 + 13 + 17 has been suggested to represent a new biomarker of heart failure. Indeed, decreased levels of this BIN1 isoform, expressed as an increased score for BIN1 in blood plasma samples from heart failure patients with preserved or reduced ejection fraction, have been reported and related to these pathologies [21,22,23]. On the contrary, our data show a decreased score for BIN1 in the plasma of KO mice without any signs of HF, but with decreased heart function. This apparent contradiction may be explained, at least in part, because <7-month-old PC1-KO mice did not show any signs of heart failure yet. Moreover, our data support our previous findings in these mice, showing that BIN1 + 13 + 17 expression levels increase in cardiac tissue compared to those observed in controls Interestingly, these results suggest that any alteration in BIN1 blood plasma levels may be taken to predict future cardiac dysfunction even when typical symptoms are not yet apparent. 

Previously, we and others have demonstrated an essential role for PC1 in maintaining normal heart function in young mice [10,11]. Thus, regulation of BIN1 expression could represent part of the pathway through which PC1 maintains normal cardiac function. Future studies are needed to determine the specific mechanism by which PC1 regulates BIN1 expression in cardiac tissue. 

As mentioned, the pathway through which PC1 regulates BIN1 expression in cardiomyocytes remains unknown; however, transcriptional factors, such as c-Myc, have been implicated as suppressors of BIN1 expression [38]. Moreover, an interesting connection between PC1 and the c-Myc pathway has been described in renal cells [39]. Further studies are required to determine whether c-Myc participates in this pathway connecting PC1 to BIN1 in cardiomyocytes. 

From a translational perspective, our discovery identifying PC1 as a positive modulator of BIN1 expression uncovers a novel mechanism implicated in ADPKD patient cardiomyopathy. The specific increase in BIN1 + 13 + 17 expression observed in PC1-KO mice may represent an early compensatory mechanism that is lost later, leading to HF. Since BIN1 organizes the T-tubule membrane calcium handling microdomains required for normal muscle contraction and T-tubule remodeling occurs early in the development of HF of varying etiology, including dilated cardiomyopathy, BIN1 gene therapy could rescue cardiac function in failing hearts. These results identify potential therapeutic targets to preserve calcium homeostasis and prevent/ameliorate cardiomyopathy development.

## 4. Materials and Methods

### 4.1. Animals

All experiments adhered to the Guidelines for the Care and Use of Laboratory Animals by the USA National Institutes of Health (8th Edition, 2011) and were approved by the Institutional Ethics Review Committee of the Universidad de Chile (CBA#0997). We used male C57BL/6 control (PC1^F/F^), and cardiomyocyte-specific PC1-knockout (PC1-KO) mice, obtained as described previously [10]. Animals were kept under standard conditions until they were used. We used Sprague–Dawley rat pups (1–3 days) to isolate neonatal rat ventricular myocytes (NRVMs). 

### 4.2. Echocardiography and Morphometric Parameters

Echocardiography was performed in anesthetized mice (isoflurane, 0.5–1%), and fractional shortening, ejection fraction, and left ventricular end-diastolic and systolic dimensions were analyzed as described previously [10]. Hearts were obtained from mice anesthetized and euthanized by cervical dislocation, and morphometric parameters were recorded (heart weight/tibia length and lung wet weight/tibia length). Ventricular heart tissue samples were obtained from mice after 7 days, 9 weeks, before any detectable signs of stress (<7-month-old mice), and after stress signs (7–9-month-old mice) were apparent. Dyspnea, shallow breathing, restricted mobility, and fatigue were considered stress signs. All tissues were frozen in liquid nitrogen and maintained at −80 °C until protein or mRNA collection or fixation for histology. 

### 4.3. NRVM Culture and Transfection

NRVM cultures and transfection with a siRNA specific to PC1 (siPC1) were performed as described previously [40]. Briefly, NRVMs were digested with pancreatin (1 μg/mL, Sigma-Aldrich, St. Louis, MO, USA) and cultured in plates preloaded with gelatin (2%, Sigma-Aldrich). NRVMs were cultured for 24 h before transfecting with siPC1 (120 nM, Sigma-Aldrich). After 48 h post-transfection, NRVMs were used to obtain mRNA.

A human full-length, membrane-anchored PC1 C-terminus (FLM-PC1, gift from Thomas Weimbs, corresponding to Addgene plasmid #41567) was transduced in NRVMs, as reported before [40]. Empty vector (cytomegalovirus (CMV)) was used as control. After 48 h of transduction, BIN1 isoforms mRNA were measured.

### 4.4. Protein Extraction and Western Blot Analysis

Proteins were extracted from mouse hearts with cold T-PER buffer (Thermo Scientific, Waltham, MA, USA) in the presence of a protease and phosphatase inhibitor cocktail (Roche, Basel, Switzerland). The protein concentration was determined by the Bradford method, and proteins were separated by SDS-PAGE, transferred to PVDF membrane (Millipore Corp, Burlington, MA, USA), and immunoblotted. Two different anti-BIN1 primary antibodies were used: Abcam ab185950, which recognizes amino acids 400 through to the C-terminus, and ab95022, which recognizes amino acids 189–398. Anti-GAPDH (Sigma-Aldrich, St. Louis, MO, USA) was used as a housekeeping control. After incubation with the appropriate secondary antibody, the antigen–antibody reaction was detected by ECL (Amersham Biosciences, Amersham, UK), and bands were quantified using Image Lab software. 

### 4.5. Enrichment of Membrane Fraction

Hearts were collected as was described above. The tissue was minced with a ceramic knife in a dish placed over ice and then homogenized in a glass/glass homogenizer at 1:5 ratio (*w*/*v*) of sucrose buffer (0.3 M sucrose and 10 mM HEPES pH 7.4) supplemented with protease inhibitors (Roche). The homogenate was centrifuged at 5000× *g* and 4 °C for 15 min to remove debris. The supernatant was kept on ice, and the pellet was resuspended and centrifuged again for 15 min at the same speed and both supernatants were pooled and centrifuged at 15,000× *g* and 4 °C for 20 min. The resulting supernatant was collected, and KCl was added to a final concentration of 0.6 M. This suspension was centrifugated at 120,000× *g* and 4 °C for 80 min. The pellet, microsomes enriched in surface membrane components, was resuspended in sucrose buffer and maintained at −80 °C. The supernatant fraction was referred to as the cytosolic fraction in subsequent experiments. The protein concentration was determined by the Bradford method.

### 4.6. RNA Isolation and qRT-PCR

As was described before [40], mRNA from tissue and NRVMs samples was isolated using TRIzol Reagent (Invitrogen, Waltham, MA, USA) and chloroform. The aqueous phase was obtained, and RNA was precipitated in the presence of isopropanol. RNA was reconstituted in DNase/RNase-free ddH_2_O. A total of 150 ng RNA from each sample was collected using the iScript cDNA synthesis kit (Bio-Rad). cDNA was diluted 10-fold with ddH_2_O and used for quantitative PCR analysis (Step One Plus, Applied Biosystems, Foster City, CA, USA). The ΔΔCt method was used to calculate relative transcript abundance with specific primers [10,12].

### 4.7. Histology

Hematoxylin/eosin or Masson trichrome staining was performed on tissue samples fixed in paraformaldehyde (4%) and maintained in PBS (1×) followed by paraffin embedding. Images were obtained at 20× magnification. 

### 4.8. Transmission Electron Microscopy (TEM)

Heart tissue samples were fixed as described previously [12]. Images were obtained at 26,500× magnification using a transmission electron microscope Philips Tecnai 12 (BioTwin, Quebec, QC, Canada), equipped with capture software iTEM Olympus Soft Imaging Solutions (Windows NT 6.1). An average of 113 and 223 T-tubules were measured per tissue sample from <7-month-old and 7–9-month-old mice, respectively. T-tubule lumen area and density were calculated with ImageJ software, 2.0.0-rc-43/1-50e version [12]. The number of T-tubules was counted per area (μm^2^). 

### 4.9. Plasma Cardiac BIN Content

Total blood was obtained as a terminal procedure by exsanguination through the vena cava. Briefly, mice were anesthetized with the inhalant isoflurane (0.5–1%). Then, blood was drawn into tubes containing EDTA, and plasma was obtained by centrifugation at 2250× *g* for 20 min. Plasma samples were frozen at −20 °C until use. Cardiac BIN1, specifically the BIN1 + 13 + 17 isoform (cBIN1), was determined in plasma using a cBIN1-specific ELISA assay provided by Sarcotein Diagnostics, and results are shown as the BIN1 score, an inverse index of plasma cBIN1 obtained from the natural logarithm of the inverse of cBIN1 plasma concentration, according to the manufacturer’s protocol.

### 4.10. Statistical Analysis

The Kaplan–Meier method was used for the survival curve analysis. All data are expressed as the mean ± SEM of the indicated number (*n*) of independent experiments and evaluated for normality distribution using the Shapiro–Wilk and Kolmogorov–Smirnov tests. Data for two groups were compared using a Student’s unpaired *t* test or one-way ANOVA followed by Tukey’s post-test for multiple comparisons. Differences were considered significant at *p* < 0.05. 

## 5. Conclusions

Our results suggest that cardiac pathologies observed in ADPKD patients could emerge as a direct consequence of cardiomyocyte PC1 dysfunction. Additionally, we identified the mechanosensor PC1 is a novel regulator of BIN1 expression in cardiac tissue that could be required to maintain tubulogenesis, T-tubule ultrastructure, and normal cardiac function.

## Figures and Tables

**Figure 1 ijms-24-00667-f001:**
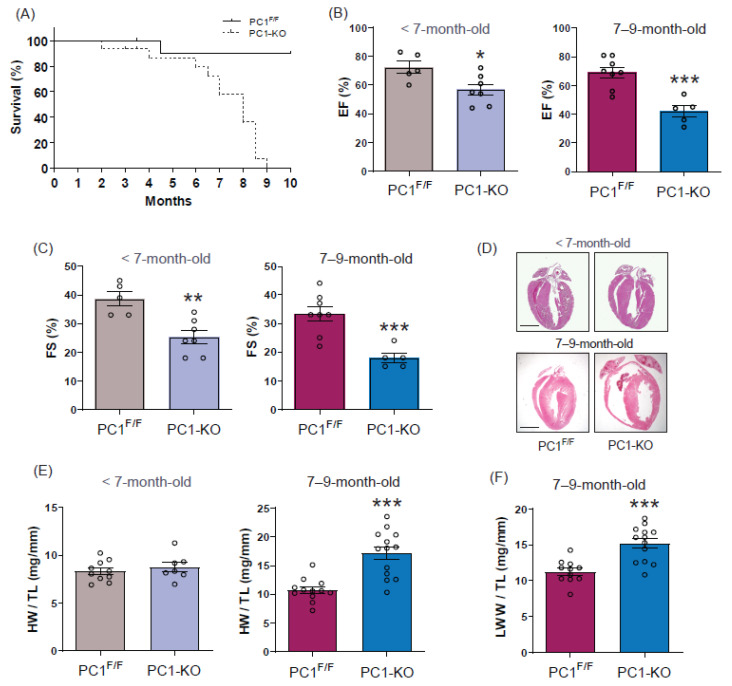
Dilated cardiomyopathy associated with loss of polycystin-1 expression. (**A**) Kaplan–Meier curves showing survival of PC1-KO mice (*n* = 12–14). (**B**) Ejection fraction (EF) and (**C**) Fractional shortening (FS) of PC1^F/F^ and PC1-KO mice without (<7 months old, *n* = 5–7) and with (7–9 months old, *n* = 5–8) signs of HF. (**D**) Four-chamber-view hematoxylin/eosin staining (Scale bar: 300 μm). (**E**) Heart weight/tibia length (HW/TL) in animals without (*n* = 7–10) or with (*n* = 12–13) signs of HF. (**F**) Lung wet weight/tibia length (LWW/TL, *n* = 11–13). Values shown are the means ± SEM and were analyzed using the Student *t* test. * *p* < 0.05; ** *p* < 0.005; *** *p* < 0.001 vs. PC1^F/F^.

**Figure 2 ijms-24-00667-f002:**
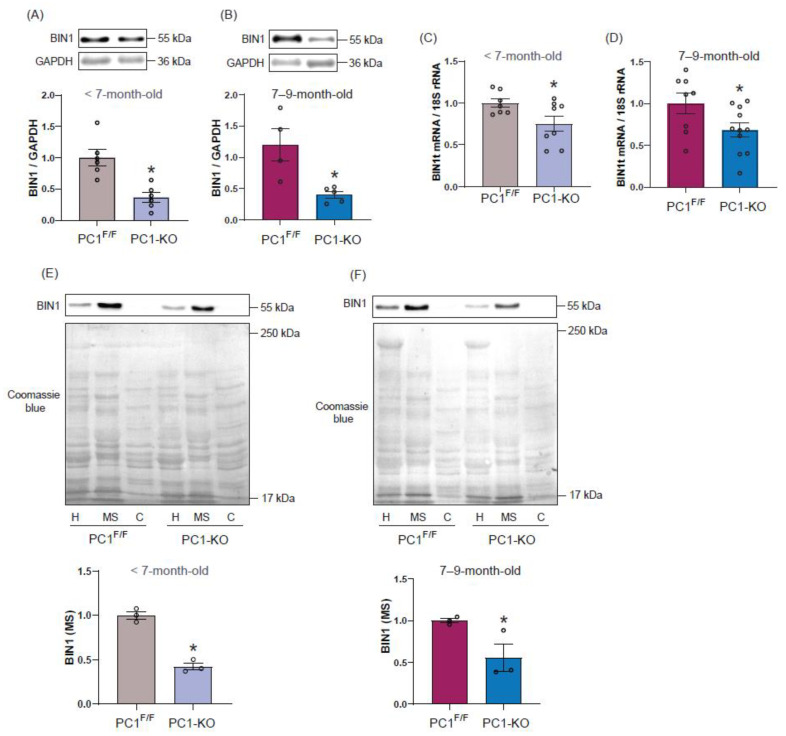
Bin1 expression and localization in cardiac tissue of PC1-KO mice. Representative Western blots and BIN1 protein quantification in samples from <7 months old (**A**, *n* = 6) and 7–9-month-old (**B**, *n* = 4–5) PC1^F/F^ and PC1-KO mice. Total BIN1 mRNA content at <7 months old (**C**, *n* = 7–8) and 7–9 months old (**D**, *n* = 8–11) in samples from PC1^F/F^ and PC1-KO mice. (**E**,**F**) Representative Western blots and quantification of BIN1 in plasma membrane-enriched fractions (*n* = 3). H: homogenate; MS: microsomes enriched in surface membrane; C: cytosolic fraction. Values shown are the means ±SEM and were analyzed using the Student *t* test. * *p* < 0.05 vs. PC1^F/F^.

**Figure 3 ijms-24-00667-f003:**
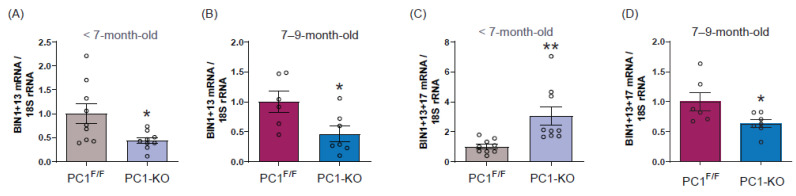
Cardiac BIN1 isoform expression is regulated by polycystin-1. BIN1 + 13 mRNA quantification in PC1^F/F^ and PC1-KO cardiac tissue samples from <7 months old (**A**, *n* = 9) and 7–9-month-old (**B**, *n* = 6–7) mice. BIN1 + 13 + 17 mRNA levels in PC1^F/F^ and PC1-KO cardiac tissue from <7 months old (**C**, *n* = 9) and 7–9-month-old (**D**, *n* = 6–7) mice. Values shown are the means ± SEM and were analyzed using the Student *t* test. * *p* < 0.05; ** *p* < 0.005 vs. PC1^F/F^.

**Figure 4 ijms-24-00667-f004:**
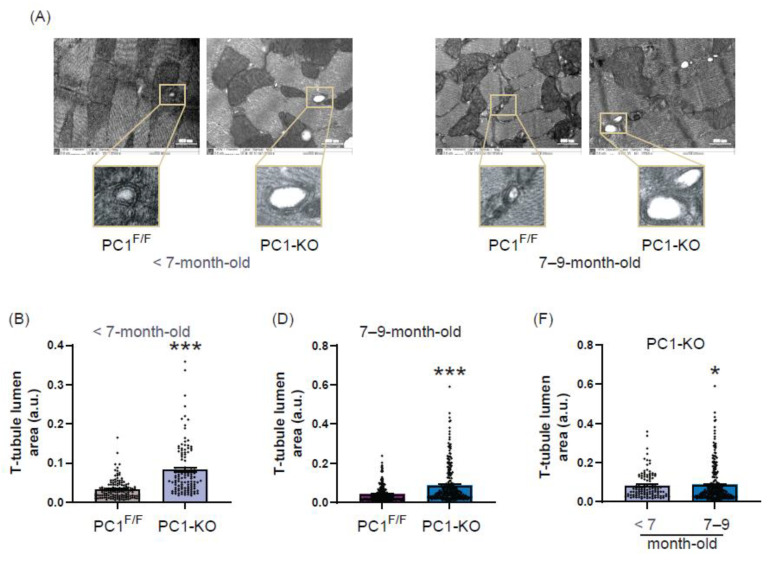
Correlation between polycystin-1 expression in cardiomyocytes and T-tubule remodeling. (**A**) Representative transmission electron microscope images of T-tubules from PC1^F/F^ and PC1-KO cardiac tissue of <7-month- and 7–9-month-old mice. In addition, the quantification of lumen area (**B**,**D**) and the lumen electron density (**C**,**E**) of T-tubules is shown. Scale bars 500 nm. (**F**,**G**) Lumen electron density of PC1-KO mice <7 months and 7–9 months of age. T-tubules were measured for 3–4 mice per genotype. Roughly 8–10 ventricular sections and 103–124 and 211–235 T-tubules were evaluated per condition in mice <7 months and 7–9 months of age, respectively. (**H**,**I**) Ratio of T-tubules number/area for cardiac tissue from PC1^F/F^ and PC1-KO mice. Values shown are the means ± SEM, and were analyzed by the Student *t* test. * *p* < 0.05; *** *p* < 0.001 vs. PC1^F/F^.

**Figure 5 ijms-24-00667-f005:**
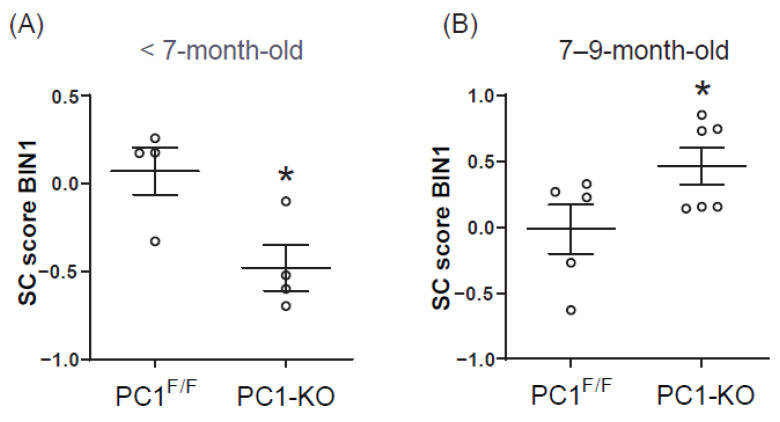
Changes in the BIN1 score of PC1-KO mice correlated with cardiac dysfunction. Bar graph of the BIN1 score obtained evaluating PC1^F/F^ and PC1-KO mice at <7 months (**A**, *n* = 4 per duplicated) and 7–9 months of age (**B**, *n*= 5–6, per duplicated). Values shown are the means ± SEM and were analyzed using the Student *t* test. * *p* < 0.05 vs. PC1^F/F^.

**Figure 6 ijms-24-00667-f006:**
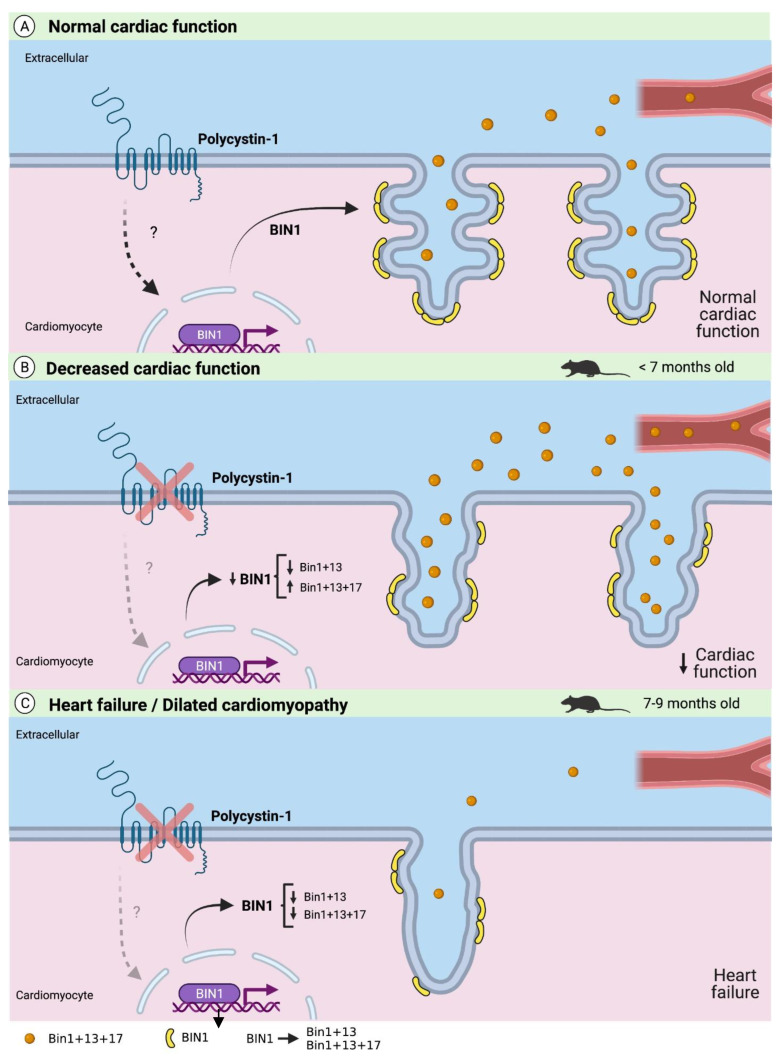
Graphical model depicting how polycystin-1-dependent BIN1 expression relates to T-tubule remodeling and cardiac dysfunction. (**A**) Normal cardiac function: PC1 positively regulates BIN1 expression and therefore T-tubule formation with microfolds. cBIN (BIN1 + 13 + 17 isoform) is released in vesicles to the extracellular space. (**B**) Decreased cardiac function without signs of HF: Total BIN1 expression and BIN1 + 13 isoform decrease in cardiospecific PC1-KO mice (<7 months) while BIN1 + 13 + 17 isoform increase leading to the loss of T-tubule microfolds and to an increase in plasma cBIN1, and; (**C**) Heart failure and development of dilated cardiomyopathy: Total BIN1, BIN1 + 13 and BIN1 + 13 + 17 expression decrease in cardiomyocytes from PC1-KO mice (7–9-month), exacerbating the loss of T-tubules and microfolds and decreasing the release of cBIN1 to the plasma. The figure was created with BioRender.com (accessed on 29 December 2021).

## Data Availability

The data presented in this study are available on request.

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
