# Peer review of "Polycystin-1 Is a Crucial Regulator of BIN1 Expression and T-Tubule Remodeling Associated with the Development of Dilated Cardiomyopathy"

_ijms, 2022, doi:10.3390/ijms24010667_

Round 1

Reviewer 1 Report

Comments to the Author

    Diaz-Vesga et al. investigated the mechanisms of onset and progression of cardiomyopathy in the settings of autosomal dominant polycystic kidney disease (ADPKD). They used cardiomyocyte-specific polycystin-1 (PC1) knockout mice and demonstrated that abnormal decrease in Bridging Integrator-1 (BIN1) and T-tubule remodeling could be the major target against cardiomyopathy in patients with ADPKD. This study challenged the important topic against idiopathic pathogenesis and provides novel findings. However, the manuscript has major concerns before acceptation for publication in IJMS. Details are listed below.

1)    Although Line 405-415 described that they need further investigation and provided some speculations, the additional investigations to clarify the relationship between PC1 and BIN1 are needed. Are there any additional results to investigate the mechanisms using cultured cardiomyocytes? In order to prove that the alterations in BIN1 was not just the consequential results of pathophysiological changes after PC1 silencing, the additional investigations are needed to confirm the direct association between PC1 and BIN1. In addition, the reason why the authors focused on BIN1 in ADPKD is not so unclear.

2)    Discussion section includes repeats of description in Introduction.

3)    The manuscript should be carefully checked and edited to avoid careless mistakes and to insert important information for readers. For example, abbreviations were used without their full spelling. There is not explanation of the definition of cBIN1 score in Figure 5 and ‘H’, ‘MS’, and ‘C’ in Figure 2.

4)    Figure 1D shows only representative photos. Results should be digitized and analyzed.

5)    Line 324-325: Figure 6 should be explained well in the text or figure legend.

Author Response

We thank both reviewers for their  critical and constructive review, which significantly improved our manuscript. We have carefully discussed their comments and revised our MS according to their suggestions. Our responses are as follow:

REVIEWER 1

Major comments:

Query 1.  Although Line 405-415 described that they need further investigation and provided some speculations, the additional investigations to clarify the relationship between PC1 and BIN1 are needed. Are there any additional results to investigate the mechanisms using cultured cardiomyocytes? In order to prove that the alterations in BIN1 was not just the consequential results of pathophysiological changes after PC1 silencing, the additional investigations are needed to confirm the direct association between PC1 and BIN1. In addition, the reason why the authors focused on BIN1 in ADPKD is not so unclear.

Response: We thank the reviewer for raising this important issue.

  1. A) To investigate if PC1 directly regulates BIN1 expression we evaluated in neonatal rat ventricular myocytes (NRVM) knockdown to PC1, the cardiospecifc isoforms of BIN1, after 48 h of transfection. In these experiments BIN1+13 mRNA decreases, while BIN1+13+17 mRNA increases, supporting our previous findings in PC1 KO mouse without signs of cardiac dysfunction (< 7-month-old mice). . The ubiquitous BIN1 isoforms (BIN1+17 and small BIN1)did not change in PC1 knockdown mice. These results suggest that BIN1 expression depends, at least in part, of PC1 expression in cardiomyocytes. These results are shown in Supplementary Figure 3 E-I and results section (3.3.).
  2. B) To clarify why we focused on BIN1 in ADPK, we added the following sentence in Introduction seccion: “Considering ADPKD patients develop cardiomyopathy and cardiospecifc PC1 KO induces cardiac dysfuntion in a mouse model ( ), we focused to determine whether BIN1 expression and T-tubule formation are dependent of PC1. This knowledge may help to understand the mechanism by which PC1 regulates cardiac function and leads to the development of cardiomyopathy in ADPK patients”.

Query 2. Discussion section includes repeats of description in Introduction.

Response: We modified the discussion section in the new manuscript version.

Query 3. The manuscript should be carefully checked and edited to avoid careless mistakes and to insert important information for readers. For example, abbreviations were used without their full spelling. There is not explanation of the definition of cBIN1 score in Figure 5 and ‘H’, ‘MS’, and ‘C’ in Figure 2.

Response: We apologize for these omissions. We checked the manuscript and add the full spelling of the abbreviations. We also explain the definition of cBIN1 score in Materials and Methods section (sub-section 2.9.). In Figure 5, we added the definition of HS, MS and C in the Figure 2 legend.

Query 4. Figure 1D shows only representative photos. Results should be digitized and analyzed.

Response: We understand the reviewer’s concern but our only purpose was  to show a qualitative image  of the four heart chambers. These kind of hematoxylin/eosin  stained images are not usually used to quantify hypertrophy .  For a more quantitative appreciation of cardiac dimentions we added data of left ventricular end-diastolic, and systolic dimensions measured by echocardiography. These data show that 7-9-month-old PC1-KO mice have increased dimensions of ventricular chambers compared to controls, however, < 7-month-old PC1-KO mice did not have differences respect to control mice. All these data are included in Supplementary Figure 1A-D, as well as in results section (3.1.).      

Query 5. Line 324-325: Figure 6 should be explained well in the text or figure legend.

Response: We apologize for this omission, we added  an explanation in the Figure 6 legend.  

Author Response

We thank both reviewers for their  critical and constructive review, which significantly improved our manuscript. We have carefully discussed their comments and revised our MS according to their suggestions. Our responses are as follow:

REVIEWER 2

Major comments:

Polycystin-1 is crucial regulator of BIN1 expression and T-tubule remodeling associated with the development of dilated cardiomyopathy Diaz-Vesga et al. The authors in this paper reported that a cardiomyocyte-specific PC1 silenced in mice lead to an impairment of cardiac function and T-tubule remodeling but no heart failure before 7 -9month of age. Then, authors correlate these phenotypes with the levels of BIN-1 expression and found interestingly that upon KO of PC1 in cardiomyocyte BIN1 levels are decreased both before 7months and between 7-9months. The paper is clearly written and the provided data are convincing. However, I have a main concern regarding the lack of rescue experiment. The provided data would be strengthened if upon re-expression of PC1, BIN1 levels were restored for instance as well as some of the PC1-KO induced effects. This could be performed even in cardiomyocyte in culture such as primary cultured widely used or cell lines.

Response: The reviewer is right and our conclusion would be strengthened by rescue experiments, however, we don´t have the tools required for in vivo rescue. Unfortunately, we cannot perform these experiments by using cultures of adult cardiomyocytes isolated from PC1-KO because our colony is small and we do not have enough mice of a similar age of the animals used in this work (aprox 3 month).  On the other hand we have no experience with cardiac cell lines and it would take us a considerable time to use these cells to perform experiments. Nevertheless, we did some experiments in neonatal rat ventricular myocytes (NRVMs) knockdown to PC1 and we evaluated BIN1 mRNA levels, to assess the direct relation between PC1 and BIN1 isoforms. Our data showed a decreased BIN1+13 mRNA and increased BIN1+13+17 mRNA after 48 h of PC1 siRNA transfection, while we did not observe significative changes in the ubiquitous BIN1 isoforms (BIN1+17 and Small BIN1). Moreover, the overexpression of PC1 full length membrane, increases all BIN1 isoforms mRNA except BIN1+13+17 mRNA. These results show that there is a direct relationship between  PC1 and BIN1 expression. All these data were included in Supplementary Figure 3 E-M and results section (3.3.).

Minor comment: Microsomal fraction enriched in plasma membrane components to monitor BIN1 membrane localization is an interesting approach although it would be much more convincing to study the localization of BIN1 directly in cardiomyocyte tissue or cells either by immunofluorescence or by TEM.

Response: We thanks the reviewer for this suggestion. However, as we said before, for any experiments with mice we would need at least 3 month-old mice, so we are unable to do these experiments at this moment.

Round 2

Reviewer 2 Report

The authors provide additional data reinforcing their findings.